# Serum Levels of miR-148b and Let-7b at Diagnosis May Have Important Impact in the Response to Treatment and Long-Term Outcome in IgA Nephropathy

**DOI:** 10.3390/jcm10091987

**Published:** 2021-05-05

**Authors:** Nikoleta M. Kouri, Maria Stangou, George Lioulios, Zoi Mitsoglou, Grazia Serino, Samantha Chiurlia, Sharon Natasha Cox, Persia Stropou, Francesco P. Schena, Aikaterini Papagianni

**Affiliations:** 1Department of Nephrology, School of Medicine, Aristotle University of Thessaloniki, General Hospital “Hippokratio”, 54642 Thessaloniki, Greece; nicol.kouri@gmail.com (N.M.K.); pter43@yahoo.gr (G.L.); zoimhts@gmail.com (Z.M.); persiags@auth.gr (P.S.); aikpapag@otenet.gr (A.P.); 2National Institute of Gastroenterology “S. de Bellis”, Research Hospital, Castellana Grotte, 70013 Bari, Italy; grazia.serino@irccsdebellis.it; 3Schena Foundation, Campus of Veterinary Medicine, University of Bari, Valenzano, 70010 Bari, Italy; samanthachiurlia@gmail.com (S.C.); sharonnatasha.cox@uniba.it (S.N.C.); paolo.schena@uniba.it (F.P.S.); 4Department of Biosciences, Biotechnology and Biopharmaceutics, University of Bari, 70121 Bari, Italy; 5Department of Emergency and Organ Transplants, University of Bari, 70121 Bari, Italy

**Keywords:** biomarker, IgA nephropathy, microRNA, serum

## Abstract

Background/aims: Previous studies showed that two microRNAs, let-7b and miR-148, which regulate the O-glycosylation process of IgA1, may predict diagnosis of primary IgA nephropathy (IgAN). The combined analysis of their serum levels in calculated statistical models may act as serum biomarkers for the diagnosis of primary IgAN. In the present study, we aimed to assess their impact not only on clinical and histological findings at onset but also on renal function after a long-term follow-up. Patients and methods: We enrolled 61 Caucasian patients with biopsy-proven IgAN. Serum levels of miR-148b, let-7b, and galactose-deficient IgA1 (Gd-IgA1) at the time of diagnosis were measured using real-time quantitative PCR and enzyme-linked immunosorbent assay using the monoclonal antibody KM55, respectively. Their values along with calculated Models 1 and 2 were correlated with histologic scoring system (Oxford classification system) and with renal function at diagnosis and after 11.9 ± 6.6 years. Fifty-five healthy volunteers were enrolled as controls. Results: No significant correlation was found between miRNA and Gd-IgA1 levels and eGFR and proteinuria at diagnosis. A significant negative association was detected between the presence of crescents and serum levels of let-7b (*p* = 0.002), miR-148b (*p* = 0.01), and Models 1 and 2 (*p* = 0.02 and *p* = 0.007, respectively). At the end of follow-up, eGFR correlated with let-7b levels (*p* = 0.01), Model 1 (*p* = 0.002), and Model 2 (*p* = 0.004). Patients with fast progression of the renal damage had significantly increased levels of let-7b (*p* = 0.01), Model 1 (*p* = 0.003), and Model 2 (*p* = 0.005) compared to slow progressors, as did those who reached ESKD (*p* = 0.002, *p* = 0.001, and *p* = 0.001, respectively). Results were most prominent in those treated with corticosteroids. Finally, cut off levels in Models 1 and 2 could also predict the renal function outcome after long-term follow-up. Conclusions: Serum levels of let-7b and miR-148b and their combination, may serve as predictors for long-term renal function outcomes, particularly in patients treated with corticosteroids.

## 1. Introduction

Immunoglobulin A nephropathy (IgAN) is the most common glomerulonephritis worldwide even though its true incidence might be underestimated since many cases are thought to remain undiagnosed. Despite many efforts to identify a specific non-invasive diagnostic biomarker, the diagnosis of IgAN still relies only on kidney biopsy, based on the characteristic mesangial IgA deposits. The clinical course of the disease is very variable ranging from stable renal function over decades to rapid decline and end-stage kidney disease (ESKD) within few years. Several attempts have been made to predict the course of the disease based on clinical [1] or histological [2] score systems but additional criteria reflecting the major players involved in the pathogenesis of IgAN could be very helpful. They could identify patients at high risk for deterioration of renal function and allow a more aggressive and targeted treatment in these cases.

A major breakthrough in the understanding of disease pathogenesis was the discovery that IgA molecules deposited in IgAN patients’ mesangium are under-glycosylated [3,4]. Galactose-deficient IgA1 (Gd-IgA1) are also detected at high levels in IgAN patient’s serum compared to healthy controls. Recently, the monoclonal antibody ΚΜ55 produced in rats immunized by a human Gd-IgA1 hinge region peptide [5] has been used as a new enzyme-linked immunosorbent assay (ELISA) to recognize Gd-IgA1. This assay is easier to perform and more reliable than the previous assays for Gd-IgA1 detection using lectins.

The underlying cause of the aberrant glycosylation of IgA1 in IgAN patients remains unknown, but a modified expression of the enzymes involved in the glycosylation process has been demonstrated [6]. The expression of these key enzymes was shown to be regulated by microRNAs (miRNAs): N-acetylgalactosaminyltransferase 2 (GALNT2) by let-7b [7] and core-1-β1,3 galactosyltransferase 1 (C1GALT1) by miR-148b [8]. Increased serum levels of let-7b and miR-148b were found in IgAN patients, giving hope that their measurement could contribute to the diagnosis of IgAN. Their combined analysis in statistical models further increased their accuracy [9].

The aim of this study was to correlate combined serum levels of miRNAs and Gd-IgA1, as measured with the new monoclonal antibody KM55, with clinical and histologic findings at onset. Furthermore, we evaluated their predictive ability on long-term renal function and on response to treatment.

## 2. Patients and Methods

### 2.1. Patients

Sixty-one adult biopsy-proven IgAN patients, a subgroup of those who had participated in the previous validation study [9], were enrolled in the present study. All patients were Greek Caucasians and had a long-term follow-up (more than 10 years). Laboratory data were collected at the time of diagnosis, before starting any treatment.

Inclusion criteria were as follows: age >18 years; diagnosis of IgAN based on renal biopsy; both optical microscopy and immunofluoresence findings; the specimen of kidney biopsy for the optical microscopy should include at least 8 glomeruli; serum sample for the evaluation of miRNAs and Gd-IgA1 levels collected at time of diagnosis and before any treatment; and finally, adequate data at diagnosis and during close follow-up, according to protocol.

Patients with secondary IgAN, due to IgA vasculitis, lupus nephritis, or liver cirrhosis, as well as patients with scarce follow-up, or those who were on immunosuppressive treatment before collection of blood samples, were excluded from the study.

Epidemiological, anthropometric, and clinical data at time of diagnosis were obtained from patients’ files. Race, sex, age, blood pressure, and biochemical results at the day of kidney biopsy, including serum creatinine, 24 h urinary protein excretion, and micro- or macro-scopic hematuria were collected. Estimated glomerular filtration rate (eGFR) was calculated using the chronic kidney disease-epidemiology collaboration (CKD-EPI) formula. The presence of systolic blood pressure (SBP) of >140 mmHg and/or diastolic blood pressure (DBP) >90 mmHg and/or use of any antihypertensive medication was defined as arterial hypertension. Fifty-five healthy volunteer blood donors, Greek-Caucasians, similar age and sex with patients, were enrolled as controls, after giving consent about the study.

The study was carried out according to the principles of the Declaration of Helsinki and was approved by the Ethics Committee of Aristotle University of Thessaloniki, Greece, code number 379, date of approval 15/9/2017. Written informed consent was obtained from all participants. Authors did not receive any founding for the present study, and there was no conflict of interest.

### 2.2. Histology

IgAN diagnosis was based on immunofluorescence findings, consisting of IgA immunoglobulin deposits in the mesangial area of glomeruli, either alone, or in co-deposition with C3 and/or IgG. Kidney biopsy specimens were scored using the histologic MEST-C “Oxford classification” system [2].

### 2.3. Measurement of let-7b and miR-148b by Real-Time PCR

Quantification of let-7b and miR-148b serum levels in patients and controls was performed by Serino G et al. [9]. In brief, RNA was extracted from the serum samples with the miRNeasy serum/plasma kit (Qiagen, Germany). The reverse transcription reaction was performed with the miScript reverse-transcription kit (Qiagen). The amounts of let-7b and miR-148b were assayed by quantitative PCR with the miScript SYBR Green PCR kit (Qiagen) using specific primers for miR-148b and let-7b (Qiagen). Normalization was performed using an internal control, the miR-27a, as demonstrated in our previous paper [9]. Serum let-7b and miR-148b relative expression were calculated using the 2-^ΔCt^ method where ΔCt = Ct _(let-7b or miR-148b)_ − Ct _miR-27a_.

### 2.4. Measurement of Galactose-Deficient IgA1 (Gd-IgA1)

Gd-IgA1 was measured in duplicate using sandwich enzyme-linked immunosorbent assay (ELISA). In brief, high-absorption polystyrene 96-microplates (Nunc MaxiSorp, Thermo Fisher Scientific, Waltham, MA, USA) were coated with 7.5 ng/mL of KM55, a specific monoclonal antibody for hinge region in human Gd-IgA1 (IBL), overnight at room temperature. After blocking with phosphate buffered saline (PBS) containing 1% bovine serum albumin (BSA) for 2 h at room temperature, samples diluted (1:400) were added in blocking buffer and standard to each well and incubated for 2 h at room temperature. After washing with PBS containing 0.05% Tween-20, the captured Gd-IgA1 was detected with horseradish peroxidase conjugated mouse antihuman IgA IgG (Southern-Biotech) diluted in blocking buffer. The reaction was developed with the peroxidase chromogenic substrate 3′,3′,5′,5′ tetramethylbenzidine (Sigma-Aldrich). The color reaction was stopped using 1N H_2_SO_4_ as stop solution, and the absorbance was measured in a microplate reader (DV 990 B/V6, GVD, Bari, Italy) at 450 nm. The concentration of Gd-IgA1 in each sample was calculated using a calibration curve constructed by the optical densities of a standard Gd-IgA1 myeloma protein. The results were expressed in ng/mL.

### 2.5. Calculation of Models Based on miRNAs and Gd-igA1 Serum Levels

The “diagnostic signatures” used in this study derived from two logistic regression models (Models 1 and 2), as described earlier [9].

Model 1: −4.415 + 0.755 _*_ ethnicity + 3.866 * log_10_(let-7b)−5.115*log_10_(miR-148b)

Model 2: −4.038 + 4.102 * log_10_(let-7b)–3.877*log_10_(miR-148)

In their first description, different models were calculated for different ethnic groups; in the present study, we adjusted the models for Caucasians, as all our IgAN patients were Greek Caucasians, and the results were used as biomarkers, including both let-7b and mi_R148 serum levels, in a combination defined previously. Efficacy of the two models to identify IgAN patients had been proven, and the cut off level of −0.19 was defined for both Model 1 and 2 to show the highest accuracy with IgAN diagnosis.

### 2.6. Follow-Up

Patients were treated according to KDIGO guidelines for primary IgAN. All patients received renin angiotensing aldosterone system (RAAS) inhibitors for at least 6 months following diagnosis and continued either with steroids, given for 6 months with gradual reduction (Pozzi regime), or with RACE inhibitors, according to the proteinuria levels [10,11].

All patients were followed-up with regularly in the outpatients’ clinic at 6 months intervals and clinical parameters, e-GFR, proteinuria, hematuria, and medication were recorded at each visit. Renal function outcome at the end of follow-up was estimated by the last eGFR levels and annual eGFR change. To determine the annual loss of eGFR, we considered the eGFR at baseline, at 6 months and subsequently, every year until the end of follow-up. The average annual change of eGFR was estimated based on the above measurements, and patients were divided into the following: slow progressors (SP): annual eGFR change ≥−1 mL/min/1.73 m^2^; moderate progressors (MP): annual eGFR change by −1 to −3 mL/min/1.73 m^2^; and fast progressors (FP): annual e-GFR change by ≤−3 mL/min/1.73 m^2^.

Based on last eGFR levels and annual changes of eGFR, we estimated patient population with ≥50% or <50% reduction of eGFR at the end of follow-up and also with patients who were SP, MP, and FP.

The primary endpoints were (i) SP, MP, or FP, based on the annual reduction rate of eGFR and (ii) ESKD. Secondary endpoint was the combination of ≥50% reduction in eGFR and ESKD.

### 2.7. Statistical Analysis

Package for social sciences (SPSS Inc., Chicago, IL, USA) for Windows, version 25.0., was used for the statistical analysis. *p* values < 0.05 (two-tailed) were considered statistically significant for all comparisons. Shapiro–Wilk and/or Kolmogorov–Smirnov tests were applied to determine normality of variables. Normal distributed continuous variables were expressed as mean ± standard deviation, while data from nonparametric variables were expressed as medians and range. Differences between groups were estimated by Student’s *t* test for paired and independent samples and ANOVA test, for normally distributed variables, and Mann–Whitney U test, Wilcoxon signed ranks test, and Kruskal–Wallis H test were used for nonparametric variables.

## 3. Results

### 3.1. Patients

The Greek Caucasian patients with primary IgAN enrolled in the present study had a mean eGFR of 60.9 ± 24.6 mL/min/1.73 m^2^ and a mean proteinuria of 1.7 ± 0.9 g/24 h at the time of diagnosis. Clinical data, histology findings of the renal biopsy, biochemical results at the time of diagnosis, and the last follow-up are reported in Table 1. The mean follow-up was 143.8 (9–301) months. All patients who did not progress to ESRD had a follow-up of more than 8 years.

Forty-one patients had hypertension at diagnosis, and 32 were already on renin-angiotensin-aldosterone system (RAAS) inhibitors and/or other anti-hypertensive agents.

According to the Oxford classification, 52 patients had M1 lesions, 16 E1, 41 S1, 2 T2, 16 T1, 47 C0, 12 C1, and 2 C2. Figure 1 depicts representative cases of the cohort of our IgAN patients

### 3.2. Serum Levels of Gd-IgA1, miR-148, and Let-7b at Onset

The serum levels of Gd-IgA1 and the two miRNAs, let-7b and miR-148b, together with the results from calculated Models 1 and 2, in 61 IgAN patients and differences with healthy controls are depicted at Table 2. Statistically significant differences were found for Gd-IgA1 levels in IgAN patients compared to 55 healthy controls (median, range) 1.16(0.3–4.9) vs. 0.73(0.06–2.31), respectively, *p* < 0.004, let-7b levels 4.14(0.32–53.8) vs. 1.65(0.31–18.37), respectively, *p* = 0.002, Model −1, 0.5[(−3.3)–5.4] vs. −1.59[(−3.9)–2.3], respectively, *p* = 0.001, and Model 2, −0.36[(−3.6)–5.3] vs. −1.6[(−3.8)–2.3], respectively, *p* = 0.001.

### 3.3. Correlation of Gd-IgA1 and miRNAs with Renal Function at Onset

No significant correlation was found between serum levels of Gd-IgA1, miR-148b, let-7b, and also Models 1 and 2 and age, SBP, DBP, eGFR, and proteinuria at time of diagnosis. However, IgAN patients with macroscopic hematuria (MH) had slightly increased levels of miR-148b and significantly increased levels of let-7b. Both models, 1 and 2, were significantly higher in the presence of MH (Table 3).

### 3.4. Correlation of Gd-IgA1 and miRNAs Levels with Histology

No correlation was found between the presence of mesangial hypercellularity, endothelial proliferation, glomerular sclerosis, and the severity of tubular atrophy with Gd-IgA1 and serum miRNAs levels. However, compared to patients with C0, those with C1 or C2 MEST-C score had significantly reduced levels of miR-148b [0.46 (0.1–2.1) vs. 0.27 (0.2–0.8), *p* = 0.01], let-7b [6.06 (0.3–53.8) vs. 1.11 (0.4- + 10.1), *p* = 0.002] and calculated Model 1 [−0.17 [(−3.3)–5.4] vs. −1.3 [(−2.9)–1.3], *p* = 0.02] and Model 2 [0.08 [(−3.6)–5.3] vs. −1.5 [(−3.4)–1.4], *p* = 0.006] (Figure 2). Gd-IgA1 serum levels had no significant differences according to the presence of crescents, C0 vs. C1+C2 (data not shown).

### 3.5. Renal Function Outcome

Mean levels of eGFR at the end of follow-up were 41.27 ± 30.3 mL/min/1.73 m^2^, with a mean annual reduction rate in eGFR of −2.9 ± 5.1 mL/min/1.73 m^2^, and Uprot levels were 1.08 ± 0.9 g/24 h. According to annual loss of renal function, 15 patients were classified as SP, 24 as MP, and 22 as FP. Differences between SP, MP, and FP regarding Gd-IgA1 and miRNA serum levels and calculated Models are shown at Table 4; significant differences were noticed only in let-7b serum levels, and Models 1 and 2 (Figure 3).

Nineteen patients progressed to ESRD; two of them died. Twenty-nine patients reached the secondary endpoint, with combined ≥50% reduction in eGFR or ESRD. Again, let-7b and Models 1 and 2 were found to have a significant impact. Results are illustrated in Figure 4.

### 3.6. The Different Effects on Steroid and RAAS Inhibitor Treatment

After the 6 month course with RAAS inhibitors, 49 patients continued with RAAS inhibitors only, 34 switched to corticosteroids, 27 patients were treated with combination of RAAS inhibitors and corticosteroids, and 14 with combination of corticosteroids and azathioprine or mycophenolate mofetil.

Patients treated with steroids had a mean eGFR of 57.5 ± 23.6 mL/min/1.73 m^2^ at the end of follow-up, which had significant negative correlation with let-7b levels (*r* = −0.5, *p* = 0.01), Model 1 (*r* = −0.5, *p* = 0.009), and Model 2 (*r* = −0.46, *p* = 0.01), but not with Gd-IgA1 or miR-148. Similarly, there were significant differences between SP, MP, and FP in let-7b serum levels, mean rank of 14.43, 12.71, and 24.31, respectively, *p* = 0.007, Model 1, mean rank of 15.57, 12.36, and 24.08, respectively, *p* = 0.008 and Model 2, mean rank of 15.29, 12.36, and 24.23, respectively, *p* = 0.007. Similarly, patients on corticosteroids, who progressed to ESRD had increased let-7b serum levels compared to those who did not progress, mean rank of 14.05 and 22.5, respectively, *p* = 0.04, Model 1, mean rank of 14.74 and 22.27, respectively, *p* = 0.01 and Model 2, mean rank of 14.87 and 23, respectively, *p* = 0.02. Patients who reached the secondary endpoint, with a combined reduction in eGFR and ESRD had increased Model 1 levels, mean rank of 13.4 vs. 21.5, *p* = 0.01 and Model 2, mean rank of 13.9 vs. 21, compared to those who did not, *p* = 0.03. No difference was evident regarding Gd-IgA and miR-148b serum levels.

Cut-off levels in both Models 1 and 2 could strongly predict renal function outcome, as defined by primary endpoints, of annual rates of renal deterioration and ESRD, and secondary endpoint of combined of >50% reduction in eGFR and ESKD. Results are shown in Table 5. All measurements had significant impact in patients treated with corticosteroids, while their importance was considerably reduced in patients treated with RAAS inhibitors only.

## 4. Discussion

In the present study we evaluated the possible correlation between serum Gd-IgA1 levels, together with two miRNAs, which regulate the glycosylation process of IgA1 molecule, miR-148, and let-7b, with clinical and kidney biopsy findings at presentation. Furthermore, we assessed their predictive ability on disease outcome after long-term follow-up. The present study is based on previously estimated miRNA serum levels, and calculated models based on the above measurements, which established their significant role in the diagnosis of primary IgAN [9].

The discovery of Gd-IgA1 was the major breakthrough in delineating the pathogenesis of IgAN. The aberrant glycosylated IgA1 molecules lead to the formation of anti-Gd-IgA1 autoantibodies and immune complexes that, deposited in glomeruli, cause mesangial proliferation and complement activation and inflammatory infiltration [12]. Serum levels of Gd-IgA1 measured by the lectin-based ELISA were found to be elevated in IgAN patients compared to healthy controls. However, the sensitivity and the specificity (76.5% and 94%, respectively) of this test were not high enough to allow the diagnosis of IgAN without the need of a kidney biopsy [13]. Furthermore, the measurement of Gd-IgA1 with a lectin-based ELISA assay was a progress, compared to previously used mass spectrometry but still had major disadvantages that prevented its widespread clinical use. Recently, ΚΜ55, a monoclonal antibody that recognizes the same epitopes on the hinge region of Gd-IgA1 was presented (5). This is a more reliable test that could be used in the clinical practice. Furthermore, KM55 has already been used in order to distinguish primary IgAN from secondary deposition of mesangial IgA in other diseases [14,15]. There is a limited experience with the use of KM55 for the measurement of serum Gd-IgA1 levels [16,17], and to our knowledge, the present study is the first report of its use in a Caucasian population. In our study, we confirmed that serum Gd-IgA1 levels are increased in IgAN patients compared to healthy controls. Furthermore, in concordance to the previous reports in Asian patients [16,17], we found no significant correlation of serum Gd-IgA1 levels, measured by KM55, with the histological features, clinical or biochemical baseline characteristics or outcome of the renal function. This failure to correlate Gd-IgA1 serum levels with disease severity and disease outcome does not eliminate its importance in early diagnosis and screening of primary IgAN but rather supports the “multi-hit” hypothesis in the pathogenesis of the disease. Gd-IgA1, antibodies, and immune-complexes are essential for the onset of IgAN, but development and progression of lesions in renal damage and outcome of renal function may be further determined by additional factors [18]. Ongoing inflammatory reactions localized to renal tissue, as has been described earlier [19,20,21,22], may be one explanation, but this process needs to be constantly upregulated by external stimulatory factors, and Gd-IgA1 molecules could potentially have major impact on that. This hypothesis, which in our opinion, needs to be elucidated further by serial measurements of Gd-IgA1, and regulatory miRNAs, could explain the pathogenesis of IgAN and assess therapeutic interventions.

Decreased expression of the enzymes required for the glycosylation has been reported in IgAN, and this fact has been in turn attributed to the overexpression of certain miRNAs that regulate these enzymes: the expression of N-acetylgalactosaminyltransferase 2 (GALNT2, first step of glycosylation) is influenced by let-7b [6,7] (first step of glycosylation), while the expression core-1-β1,3 galactosyltransferase 1 (C1GALT1, second step of glycosylation) is regulated by miR-148b [8].

miRNAs are small single-stranded noncoding RNA molecules. They regulate the gene expression on post-transcriptional level. They pair with complementary sequences of the target mRNA molecules repressing their translation into proteins in the ribosomes. A given miRNA may target several different mRNAs, and on the other hand, a given mRNA might be regulated by multiple miRNAs. This might explain part of the complexity of multifactorial diseases and helps us understand why a single factor is not exclusively 100% responsible for a clinical phenotype in these cases. Starting with chronic lymphocytic leukemia, altered expression of miRNAs has been associated with development of several forms of cancer. miRNAs are thought to play a dual role both in suppressing or promoting oncogenic potential [23]. Several miRNAs have been implicated in the pathogenesis of chronic kidney diseases, especially in hypertensive nephropathy, diabetic nephropathy, and tubulointerstitial fibrosis, giving hope that they could be used for diagnostic or even therapeutic purposes in the future [24]. Modified expression of miRNAs has been documented also in different forms of glomerulonephritis including IgAN (reviewed in Nalewajska M et al. [25]). However, of all miRNAs investigated in IgAN, only miR-148b and let-7b are involved in the glycosylation process and thus in the pathogenesis of IgAN reducing the possibility that any reported association is spurious. In the previous study, Serino et al. proved that the combination of miR-148b and let-7b, also taking into account patients ethnicity, resulted in two logistic regression models, 1 and 2, which dramatically improved the diagnostic efficiency of the two molecules when examined separately. It means that the two models could be used as “combined biomarkers” for the diagnosis of primary IgAN, increasing AUC to 0.82. Based on these impressive results, the question that subsequently raised were whether these molecules, separately, or in combination, had the efficacy to predict clinical presentation, histology, and renal function outcome.

In the present study, we confirmed the over expression of let-7b in IgAN patients compared to healthy controls, but there was a discrepancy regarding miR-148b serum levels. We did not find a significant increase of miR-148b in our IgAN patients, and we anticipate that our present results, together with the conflicting findings of the two previous reports (elevated levels of miR-148b in plasma [8] but decreased in serum [9] of IgAN patients compared to healthy controls) indicate that miR-148b could have a weaker impact than let-7b, at least in our Greek Caucasian patients. We found no association of let-7b and miR-148b with clinical or laboratory findings at the time of renal biopsy (apart from the presence of macroscopic hematuria). This is in concordance with the previous reported lack of a similar association of the serum Gd-IgA1 levels with clinical or laboratory features at presentation. Interestingly, we found a decreased expression of both let-7b and miR-148b in the presence of crescents (C1 or C2 of the MEST-C score) in IgAN. This may point to different pathways involved in the pathogenesis of crescentic IgAN, where the level of underglycosylation might be of secondary importance. Previous studies have described differences in the pathogenesis, histology, and presentation between IgAN and IgA vasculitis; regarding the pattern of glomerular deposits, the impact of systemic inflammation, role of complement activation, and disease outcome supporting the above suggestion [26,27,28].

The major finding of our study is the association of let-7b (and calculated Models 1 and 2) with renal function after long-term follow-up. This association could be verified by several approaches: (a) classification of the patients in slow, moderate, and fast progressors, based on the average annual change of eGFR, (b) ≥50% reduction of eGFR at the end of follow-up compared to baseline, and (c) reaching of ESRD, all giving similar results making our observation more reliable. Intriguing is the finding that the impact of let-7b and calculated Models 1 and 2 on the outcome of renal function was more prominent in patients treated with corticosteroids than those treated with RAAS inhibition alone. The present was not a randomized study, and the decision to treat patients with steroids was taken by the physicians based on the standard indications that reflect a potential for a more rapid decline of renal function. Among those, clinically identified as high-risk patients, let-7b and calculated Models 1 and 2 could identify patients who would progress despite the additional therapy with immunosuppressive agents, i.e., those resistant even to aggressive treatment.

The major strong point of the present study is the fact that all samples were collected at the time of renal biopsy and that there was a long-term follow-up (mean of approximately 12 years). Furthermore, our cohort was homogenous with the same ethnic origin, and all patients were submitted to the same treatment protocols. We trust that these advantages overcome the limitations that are derived from the relatively small sample size. According to our results, let-7b serum levels seemed to play the most important role in predicting renal function outcome of IgAN patients. Although previous studies have shown the significant role of several miRNAs in the development and progression of different forms of glomerulonephritis and outcome of kidney disease, the pathogenetic mechanisms are usually uncertain [25,29,30].

In our study, we anticipate that this effect of let-7b in renal function long term outcome is maintained by sustained upregulation of Gd-IgA1 molecules, and as mentioned above, it would be of interest to prospectively evaluate serum Gd-IgA1 and let-7b levels in IgAN patients and estimate the effect of different treatment strategies.

In conclusion, we report that, in a group of Greek Caucasian IgAN patients, serum levels of miRNAs let-7b and miR-148b, and Gd-IgA1, as well as their combined analysis not only could assist in the diagnosis of IgAN, but could also serve as predictive markers of renal function outcome and response to treatment.

## Figures and Tables

**Figure 1 jcm-10-01987-f001:**
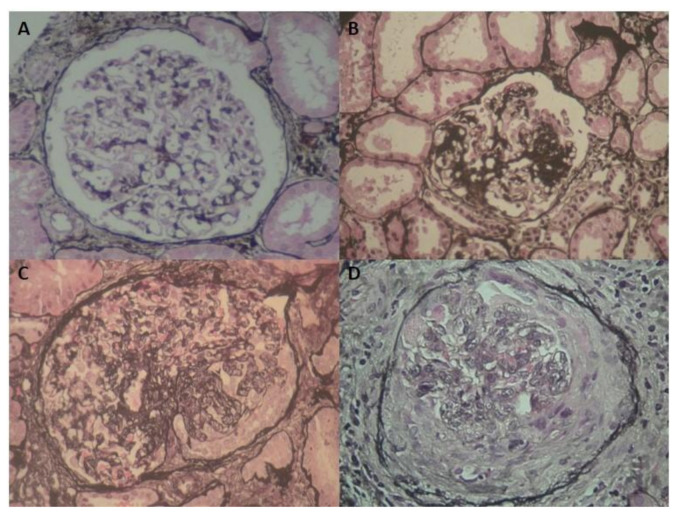
Histology of IgAN patients showing glomeruli with mesangial hyperplasia (**A**), focal segmental sclerosis (**B**), endocapillary and extracapillary hypercellularity (**C**), and cellular crescent (**D**).

**Figure 2 jcm-10-01987-f002:**
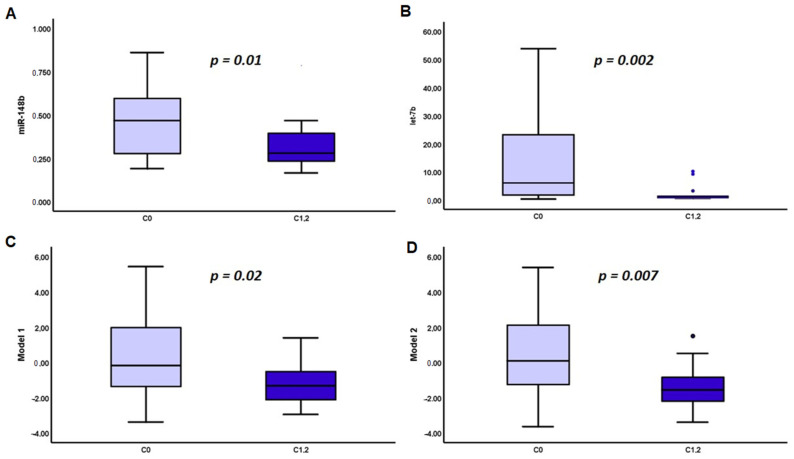
Serum levels of miR-148 (**A**), let-7b (**B**) at time of renal biopsy, Model 1 (**C**), and Model 2 (**D**) in IgAN patients with MEST-C0 and MEST-C1,2.

**Figure 3 jcm-10-01987-f003:**
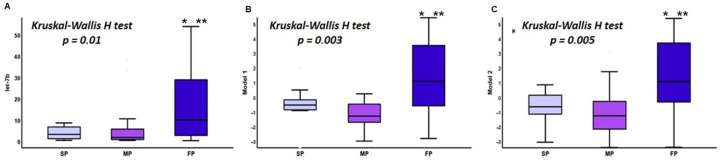
Differences in serum levels of let-7b (**A**), Model 1 (**B**), and Model 2 (**C**) between SP, MP and FP. * *p* < 0.001 (FP vs. SP), ** *p* < 0.01 (FP vs. MP). SP: Slow Progressors, MP: Moderate Progressors, FP: Fast Pogressors.

**Figure 4 jcm-10-01987-f004:**
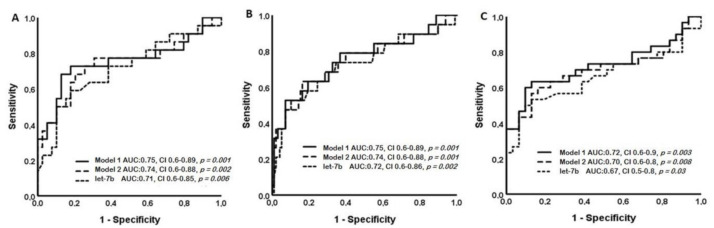
Impact of mRNA let-7b serum levels and calculated models in the outcome of renal function, as defined by fast progression (**A**), ESRD (**B**), and combined ≥50% reduction in eGFR and ESRD (**C**).

**Table 1 jcm-10-01987-t001:** Clinical data, laboratory, and histological findings of IgAN patients at time of diagnosis and last follow-up.

	IgAN	Healthy Controls
*n*	61	55
Age (years) (Median, range)	41.2 (20–66.8)	43.2 (20.5–64)
Gender (female), *n*(%)	24 (39.34%)	22 (40%)
eGFR (CKD-EPI) (ml/min/1.73 m2) (M ± SD)	60.9 ± 24.6	n.d.
Uprot (mg/24 h) (M ± SD)	1.7 ± 0.9	n.d.
Microhematouria (mH) *n*(%)	60 (98.36%)	n.d.
Macrohematuria (MH) *n*(%)	15 (24.59%)	n.d.
SBP (mmHg) (M ± SD)	145 ± 12	125 ± 10
DBP (mmHg) (M ± SD)	98 ± 5	85 ± 4
Renal Biopsy		
M0/M1	9/52	-
E0/E1	45/16	-
S0/S1	20/41	-
T0/T1/T2	43/16/2	-
C0/C1/C2	47/12/2	-
End of follow-up		
Follow-up (years) (median, range)	11.9 (0.7–25.1)	n.d.
eGFR (CKD-EPI) (ml/min/1.73 m2) (M ± SD)	41.27 ± 30.3	n.d.
Uprot (mg/24 h) (M ± SD)	1.08 ± 0.9	n.d.
≥50% eGFR reduction + ESRD *n*(%)	29 (47.54%)	n.d.
Annual change of eGFR (M ± SD)	−2.9 ± 5.1	n.d.
SP/MP/FP	15/24/22	n.d.

Abbreviations: IgAN: IgA Nephropathy, eGFR: estimated Glomerular Filtration Rate, SBP: Systolic Blood Pressure, DBP: Diastolic Blood Pressure. CKD-EPI: Chronic Kidney Disease Epidemiology Collaboration, M ± SD: Mean ± Standard Deviation, SP: Slow Progressors, MP: Moderate Progressors, FP: Fast Pogressors, n.d.: not defined.

**Table 2 jcm-10-01987-t002:** Serum levels of Gd-IgA1, miR-148, and let-7b at time of renal biopsy, and calculated models in IgAN patients and healthy controls.

	IgAN	Healthy Controls	*p*
*n*	61	55	
Gd-IgA1 *	1.16(0.3–4.9)	0.73(0.06–2.31)	0.004
miR-148b	0.42(0.16–2.14)	0.4(0.12–1.5)	NS
let-7b	4.14(0.32–53.8)	1.65(0.31–18.37)	0.002
Model 1	−0.5[(−3.3)–5.4]	−1.59[(−3.9)–2.3]	0.001
Model 2	−0.36[(−3.6)–5.3]	−1.6[(−3.8)–2.3]	0.001

* The levels of Gd-IgA1, miR-148b, let-7b, and Models 1 and 2, in patients and controls, are expressed as median and range. NS: Not Significant.

**Table 3 jcm-10-01987-t003:** Differences in Gd-IgA1, miR-148, and let-7b serum levels at time of renal biopsy, and calculated models between patients with and without MH.

	MH (–)	MH (+)	*p*
	*n* = 46	*n* = 15	
Gd-IgA1 *	1.23(0.4–4.8)	1.28(0.3–4.2)	NS
miR-148b	0.39(0.1–0.9)	0.51(0.2–2.1)	0.06
let-7b	1.51(0.3–46.8)	25.9(0.4–53.8)	<0.0001
Model 1	−0.89[(−3.3)–5.4)	1.7[(−2.9)–5.2]	<0.0001
Model 2	−1.12[(−3.6)–5.3]	2.08[(−3.4)–5.2]	<0.0001

* The levels of Gd-IgA1, miR-148b, and let-7b and results from Models 1 and 2, in patients and controls, are expressed as median and range. NS: Not Significant.

**Table 4 jcm-10-01987-t004:** Differences in serum concentration of Gd-IgA1, miR-148, and let-7b and in Models 1 and 2 between patients with slow, moderate, and fast progression (SP, MP and FP, respectively).

	SP	MP	FP	*p*
*n*	15	24	22	
Gd-IgA *	1.55(0.9–4.8)	1.28(0.4–4.2)	1.1(0.3–4.1)	NS
miR-148b	0.5(0.1–2.1)	0.43(0.1–0.48)	0.39(0.2–0.8)	NS
let7b	3.24(0.4–32)	1.7(0.4–38)	10(0.3–53.8)	0.01
Model 1	−0.52[(−3.3)–1.9]	−1.28[(−2.9)–3.1]	1.1[(−2.8)–5.4]	0.003
Model 2	−0.64[(−3.6)–2.4]	−1.26[(−3.4)–3.4]	1.07[(−3.4)–5.3]	0.005

* The levels of Gd-IgA1, miR-148b, let-7b, and Models 1 and 2, in patients and controls, are expressed as median and range. NS: Not Significant.

**Table 5 jcm-10-01987-t005:** Cut-off levels in both Models 1 and 2 had significant impact in renal function outcome, as this was defined by the primary endpoint of slow, moderate, and fast progression, and secondary endpoints of ESRD and combined ≥50% reduction in eGFR and ESRD.

	Annual rate of progression (SP, MP, FP)			Combined ≥50% reduction and ESRD			ESRD		
	OR	95% CI	*p*	OR	95% CI	*p*	OR	95% CI	*p*
	**All Patients**								
Model 1 (cutoff levels −0.19)	1.1	0.94–1.3	<0.0001	10.1	2.8–36	<0.0001	4.6	1.4–14.5	0.007
Model 2 (cutoff levels −0.19)	1.1	0.93–1.3	<0.0001	4.9	1.6–14.8	0.003	3.5	1.1–11	0.02
	**Patients treated with steroids**								
Model 1 (cutoff levels −0.19)	1.1	0.58–1.06	<0.0001	22.8	2.4-−214	0.001	8.3	1.6–42	0.007
Model 2 (cutoff levels −0.19)	0.7	0.2–1.2	0.006	6.6	1.3–32	0.01	4.9	1.2–12.3	0.03
	**Patients treated with RAASi**								
Model 1 (cutoff levels −0.19)	0.4	0.2–1	NS	5.8	1–32	0.03	1.7	0.3–9.1	NS
Model 2 (cutoff levels −0.19)	0.2	−0.4–0.8	NS	4	0.8–20	NS	1.8	0.3–10	NS

## Data Availability

The data presented in this study are available on request from the corresponding author. The data are not publicly available due to confidentiality.

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
