# Peer review of "Serum Levels of miR-148b and Let-7b at Diagnosis May Have Important Impact in the Response to Treatment and Long-Term Outcome in IgA Nephropathy"

_jcm, 2021, doi:10.3390/jcm10091987_

Round 1
Reviewer 1 Report
- Please show on the figures the statistically significance differences and indicate between which groups there is a difference.
- The references style should be adapted to the editorial requirements.
- Please discuss in more details, use more references.
- Please show the histology results.
- Please give the information about Funding, author contributions, conflict of interest, institutional approval and written constent from each patients.
- Some phrases, like "In the present study,we confirmed the over expression of let-7b in IgAN patients 320compared to healthy controls, but we could not do the same for miR-148b" are not support by the obtained results and should be rewritten.
- Please present inclusion and exclusion criteria.
- Who was a control group?
Author Response
Responses to reviewer 1.
Point 1. Please show on the figures the statistically significance differences and indicate between which groups there is a difference
Response:Thank you very much for this comment, we have shown differences on the figures, indicated statistical significance between groups
Point 2. The references style should be adapted to the editorial requirements.
Response: Thank you very much, we have adapted the refences style
Point 3. Please discuss in more details, use more references.
Response: Thank you for your suggestion. We have expanded the discussion and used more references
Point 4. Please show the histology results.
Response: Thank you for this comment. WE have added a figure of representative histologic findings of our IgAN patients in optical microscopy. We also show their MEST-C scores on table 1 and in paragraph 3.1 of the Results section
Point 5. Please give the information about Funding, author contributions, conflict of interest, institutional approval and written constent from each patients.
Thank you very much for this comment, you are absolutely right about this. Information about institutional approval, funding, patients written consistent and conflict of interest are given at the end of paragraph 2.1, at Patients and Methods section. Author contributions are analysed in a different section, according to journal guidelines.
Point 6. Some phrases, like "In the present study,we confirmed the over expression of let-7b in IgAN patients compared to healthy controls, but we could not do the same for miR-148b" are not support by the obtained results and should be rewritten.
Thank you for this suggestion. We did not find any significant difference in the serum levels of miR-148b in our Greek Caucasian IgAN patients, although there was a significant increase in let-7b levels. We have changed the sentence, according to eviewers suggestions, and make it more specific.
Point 6. Please present inclusion and exclusion criteria.
Thank you very much, you are absolutely right. We have added inclusion and exclusion criteria in paragraph 2.1, at Patients and Methods section
7. Who was a control group?
In the same section Patients and Methods, paragraph 2.1, we give some information about the control group, they were healthy volunteers of Greek-Caucasian origin, blood donors, similar age and sex with patients, and they were consented about the study
Reviewer 2 Report
IgA nephropathy is the most frequent primary glomerulonephritis worldwide, leading eventually to end-stage renal disease. Due to its heterogeneity, it is of utmost interest to find potential both laboratory and clinical parameters predicting the disease outcome.
- I do not fully understand the idea of models derived from logistic regression. Please clarify it. Authors should present this idea in more understandable manner, because it is crucial to understand the results.
- “eGFR at last follow up correlated significantly with let-7b serum levels (r=- 0.33, p=0.01), Model 1 (r=-0.42, p=0.002) and Model 2 (r=-0.39, p=0.004).”
Although correlations were statistically significant, with p<0.05, I am not convinced, that in this work, they are really vital. Correlation strength is described by “r”. In the presented paper they vary from -0.42 to -0.33 and therefore should not be treated as significant. They show rather week correlation although statistically significant.
- “Patients treated with steroids had a mean eGFR of 57.5±23.6ml/min/1.73m2 at the end of follow up, which had significant negative correlation with let-7b levels (r=-0.5, p=0.01), Model 1 (r=-0.5, p=0.009) and Model 2 (r=-0.46, p=0.01), but not with Gd-IgA1 or miR-148.
What were the criteria for starting steroid treatment?
- “Furthermore, we assessed their predictive ability on disease outcome after long-term follow up.”
Although Authors write about prediction, they do not use any statistical test for prediction.
Author Response
Responses to Reviewer 2
Point 1. I do not fully understand the idea of models derived from logistic regression. Please clarify it. Authors should present this idea in more understandable manner, because it is crucial to understand the results.
Response. Thank you for this comment, we analyse the use and apply of the two models in the section of Discussion. In the forst place, logistic regression models were clculated based on both miRNA levels, because both miRNAs have direct impact and regulate glycosylation of IgA1 molecule. There act on different stages of glycosylation, and therefore, it seems that there is an interaction between them.
Point 2. eGFR at last follow up correlated significantly with let-7b serum levels (r=- 0.33, p=0.01), Model 1 (r=-0.42, p=0.002) and Model 2 (r=-0.39, p=0.004).”
Although correlations were statistically significant, with p<0.05, I am not convinced, that in this work, they are really vital. Correlation strength is described by “r”. In the presented paper they vary from -0.42 to -0.33 and therefore should not be treated as significant. They show rather week correlation although statistically significant.
Response. Thank you for the comment. We also noticed the low levels of Rs, however, as we are talking about biomarkers, and there is a statistical significance, we think there is a potential interest in their role.
Point 3. “Patients treated with steroids had a mean eGFR of 57.5±23.6ml/min/1.73m2 at the end of follow up, which had significant negative correlation with let-7b levels (r=-0.5, p=0.01), Model 1 (r=-0.5, p=0.009) and Model 2 (r=-0.46, p=0.01), but not with Gd-IgA1 or miR-148.
What were the criteria for starting steroid treatment?
Response. Patients were treated according to KDIGO guidelines, starting with RAAS inhibitors for 6 months, and continuing with steroids if proteinuria levels were not reduced to <1gr/24hr.
Point 4. Furthermore, we assessed their predictive ability on disease outcome after long-term follow up.”
Although Authors write about prediction, they do not use any statistical test for prediction.
Response. Thank you for your comment. We evaluated the impact of two miRNAs and the calculated models, in the outcome of renal function, by estimating differences in their serum levels between patients who progressed or not to renal impairment. We believe that the relatively small number of patients does not allow us to proceed to more advanced statistical methodology
Round 2
Reviewer 1 Report
The authors responded to my comments.
Thank you !
Author Response
Thank you
Reviewer 2 Report
- The idea of regression models are stil not explained. What are they predicting? Logistic regression is used for finding correlation between numeric data and a dichotomous variable.
- Correlation strength is low and it does not matter that they are statistically significant. Maybe, chosen model is inappropriate? It is presented just a simple linear correlation, supposably a regression model or even non linear model should be applied?
- The title is still about prediction although no prediction statistic was applied.
Author Response
Point 1. The idea of regression models are stil not explained. What are they predicting? Logistic regression is used for finding correlation between numeric data and a dichotomous variable.
Response: Thank you for your comment. I fully appreciate your doubts about using a logistic regression model, I just want to clarify that these models were previously calculated and their efficacy to diagnose primary IgA Nephropathy was published in the following paper: Kidney Int. 2016;89:683-692
In a retrospective International study circilating miR-148b and let-7b were found to be serum markers for detecting primary IgA Nephropathy.In the present paper, we used serum levels of miR-148b, let-7b and their combination, as "biomarkers" and evaluated their impact to histology and response to treatment
Point 2. Correlation strength is low and it does not matter that they are statistically significant. Maybe, chosen model is inappropriate? It is presented just a simple linear correlation, supposably a regression model or even non linear model should be applied?
Response: Thank you for the comment. We erased the correlation of our biomarkers with eGFR levels at last follow up, in fact, we estimated outcome of renal function based on primary and secondary end points
The title is still about prediction although no prediction statistic was applied.
Response: We have made changes in the title